# New Insights into the Metabolic Profile and Cytotoxic Activity of *Kigelia africana* Stem Bark

**DOI:** 10.3390/molecules30061388

**Published:** 2025-03-20

**Authors:** Dimitrina Zheleva-Dimitrova, Rositsa Mihaylova, Maria Nikolova, Nisha Singh, Spiro Konstantinov

**Affiliations:** 1Department of Pharmacognosy, Faculty of Pharmacy, Medical University of Sofia, 1000 Sofia, Bulgaria; 2Department of Pharmacology, Pharmacotherapy and Toxicology, Faculty of Pharmacy, Medical University of Sofia, 1000 Sofia, Bulgaria; rmihaylova@pharmfac.mu-sofia.bg (R.M.); maria.nikolova.sofia@abv.bg (M.N.); skonstantinov@pharmfac.mu-sofia.bg (S.K.); 3School of Life Sciences, University of KwaZulu-Natal, Durban 4000, South Africa; singhni@ukzn.ac.za

**Keywords:** *Kigelia africana*, UHPLC-HRMS profiling, cytotoxic activity

## Abstract

Ultra-high-performance liquid chromatography coupled to Orbitrap high-resolution mass spectrometry (UHPLC-HRMS) was recently employed in many fields to obtain a rapid characterization of plant extracts. *Kigelia africana* (family Bignoniaceae) is a quintessential African herbal medicinal plant with immense indigenous medicinal and non-medicinal applications. The aim of the present research was to obtain an in-depth metabolite profiling of the *K. africana* stem bark extract using UHPLC-HRMS and to conduct a preliminary screening of its anticancer activity against a panel of malignant human cell lines of different origin. The UHPLC-HRMS analysis revealed 63 secondary metabolites including phenolic acids, gallo- and ellagitannins, iridoids, naphthoquinones, and anthraquinones. A total of 34 of all annotated compounds are reported for the first time in *K. africana* stem bark. The studied profile was dominated by trimethylellagic acid, dimethylellagic acid isomers, and ellagic acid. In all tumor models, we established a pronounced inhibition of cell growth in a mostly dose-dependent manner, with IC_50_ values ranging near and well below (4–30 µg/mL) the lowest treatment concentration of 25 µg/mL. The established cytotoxicity profile of the *K. africana* extract, highly biased toward malignantly transformed but not normal cells, suggests specific modulation of defined molecular tumor targets. This study revealed *K. africana* stem bark as a new source of gallo- and ellagitannins, and highlighted the studied herbal drug as an antiproliferative agent with potential pharmaceutical application.

## 1. Introduction

*Kigelia africana* (Lam.) Benth. (syn. *K. pinnata* (Jacq.) DC.) (family Bignoniaceae), commonly known as sausage tree due to the shape of its fruit, is a native of the African continent where it is commonly found in the southern, central, and western regions [1]. *K. africana* is a large tree, growing to 20 m in height, and widely used for the treatment of a wide range of diseases and conditions in addition to its use as an agroforestry plant. Notably, wild plants, including *K. africana*, hold significant potential not only for direct therapeutic approaches but also for detoxifying harmful substances (xenobiotics) [2,3]. Ethnopharmacological studies have revealed the therapeutic relevance of various preparations from different *K. africana* parts to treat a wide range of skin complications, dysentery, constipation, wounds, ulcers, gonorrhoea, rheumatism and abscesses [1].

Overall, more than 150 secondary metabolites including naphthoquinones, iridoids, flavonoids, coumarins, terpenes, terpenoids and steroids have been reported in the *K. africana* extracts. Iridoids were found to be the major chemical compounds in *K. africana*. Norbiturnial, 10-deoxyeucommiol, 7-hydroxy-10-deoxyeucommiol, 7-hydroxy eucommic acid, des-p-hydroxybenzoyl kisasaganol, jofuran, together with the iridoid glycosides ajugol, caffeoyl ajugol, catalpol, specioside, verminoside, and minecoside were previously isolated from different parts of *K. africana* [4,5,6,7,8,9]. Limonoids (kigelianolide, khayanolide B, diacetylkhayanolide E were also reported [10].

An additional class of major metabolites present in *K. africana* is the naphthoquinones. Lapachol, and its derivative dehydro-α-lapachone, kigelinone, and pinnatal were the first isolated from the root, wood, and fruit respectively [11,12,13]. Phenylethanoid glycosides from *K. africana* play a major part in its medicinal properties. Decaffeoylacteoside, darensdoside A, verbascoside, isoacteoside, echinacoside, 6-*p*-coumaroylsucrose, and others were isolated from the fruit of *K. africana* [14].

Several flavonoids were isolated from the leaves and bark of *K. africana*. Among them are quercetin, luteolin, 6-hydroxy luteolin, together with their glycosides, isovitexin from the leaves and isoschaftoside from the fruit [14]. Coumarins are among the first phytoconstituents reported for *K. africana*. 6-Methoxymellein, kigelin, and 3-demethylkigelin were isolated from the root and bark [11], while 6-demethylkigelin was found in the fruit, bark and leaves [15]. Other phenolic compounds found in *Kigelia* are 4-hydroxycinnamic acid, caffeic acid, caffeic acid methyl ester, ferulic acid, and 3, 4-dimethoxy cinnamic acids, kojic acid, melitolic acid, ethylgallic acid, chlorogenic acid, ellagic acid, and rosmarinic acid [1,16]. Previously, using a non-targeted high-performance liquid chromatography coupled to high-resolution time-of-flight (HPLC-MS/MS TOF), 103 compounds (peptides, phenolic acids, coumarins, naphthoquinones) were identified in the *K. africana* methanol and aqueous fruit extracts [5]. Compound dereplication was carried out using the METLIN metabolomics database and Forensic Toxicology, using the Find Formula function in the software package. Recently, a rapid evaporative-ionization mass spectrometry (REIMS) coupled with an electroknife led to the identification of 78 biomolecules, including phenols, fatty acids, and phospholipids in the fruits. The compound annotation was performed by Humane Metabolome Database, LipidMaps, and METLIN platforms [17]. More than 235 phytoconstituents were annotated, using UHPLC-TOF-MS in the *K. africana* fruit ethyl acetate extract, including atypical biologically active compounds, yohimbine, and psilocybin [18]. LC/MS analysis of the root bark extract of *K. pinnata* revealed the presence of 63 phytochemicals from the groups of terpenoids, flavonoids, phenols, glycosyl diterpenoids, naphthoquinones, and steroids [19].

The phytochemistry, pharmacology, and traditional medicinal applications of *K. africana* were the subject of literature reviews by Bello et al., 2016, Nabatanzi et al., 2020, and Assanti et al., 2022, which highlighted the presence of antibacterial, antifungal, analgesic, anti-inflammatory, antidiabetic, antiprotozoal antioxidant, and anticancer activities of the plant [1,20,21]. Prominent cytotoxic and antiproliferative properties against several carcinoma cell lines have been reported for *K. africana* extracts. Bark extracts have shown promising results in treating of melanoma, renal carcinoma [9], and breast cancer tumor models [15]. Furthermore, the root bark exhibited notable activity against KB cells [22]. Momekova et al., 2012 reported significant in vitro cytotoxicity of *K. africana* methanolic stem bark extract against a panel of human cancer cell lines of both leukemic and epithelial origin (DOHH-2, SKW-3, REH, MCF-7, HL-60, HDMY-Z, K-562) [23]. Potent antiproliferative activity against Caco-2 and HeLa carcinoma cell lines was demonstrated by *K. africana* methanolic fruit extracts [5]. The methanol, water, and ethyl acetate fruit extracts have also displayed significant antiproliferative activity against Jeg-3 choriocarcinoma cells [5].

Despite numerous studies on *K. africana* phytochemistry and pharmacology, no comprehensive metabolite profiling has yet been reported for the stem bark extract by means of ultra-high-performance liquid chromatography—Orbitrap high-resolution mass spectrometry (UHPLC-HRMS). Therefore, conducting a thorough investigation of all compounds present in *K. africana* stem bark is needed. Detailed information on the metabolite profile in the stem bark would provide valuable information on the biological potential of these compounds. Considering previous studies, we assumed an in-depth profiling of the secondary metabolites in *K. africana* methanol-aqueous extract using UHPLC-HRMS, combined with an assessment of its cytotoxic activity. Herein, the majority of annotated gallo- and ellagitannins were reported for the first time in *K. africana*.

## 2. Results and Discussion

### 2.1. UHPLC-HRMS Profiling of Secondary Metabolites in K. africana Stem Bark Extract

Based on the MS and MS/MS accurate masses, retention times, fragmentation patterns in MS/MS spectra, relative ion abundance, and comparison with reference standards and literature data, a total of 63 specialized natural products were identified or tentatively annotated in *K. africana* stem bark extract (Table 1). 

Identification confidence levels for metabolite profiling were performed according to Sumner et al., 2007 [24] and were as follows: level 1—compounds identified by comparison to the reference standard; level 2—putatively annotated compounds, 3—putatively characterized compound classes. Six compounds were assigned to phenolic acids and flavone. Seven compounds were ascribed as gallotannins. Thirty-one ellagitannins, five iridoids, and fourteen naphthoquinones and anthracene derivatives were also identified in the studied extract. Among all these sixty-three metabolites, one phenolic acid derivative, five gallotannins, nineteen ellagitannins, one iridoid, one naphthoquinone, and seven anthraquinones are reported for the first time in *K. africana* stem bark.

#### 2.1.1. Phenolic Acids and Flavonoids

Hydroxybenzoic acids (**1** and **3**), a pentoside (**2**), and caffeic acid (**4**) were identified based on comparison of retention times, exact masses, and fragment spectra with reference standards and literature data (Table 1) [25]. Compound **5** ([M − H]^−^ at *m*/*z* 247.025, C_12_H_8_O_6_) gave a series of low abundant ions at *m*/*z* 219.030 [M − H-CO]^−^, 191.034 [M − H-2CO]^−^, 163.039 [M − H-3CO]^−^, 135.044 [M − H-4CO]^−^, and 107.049 [M − H-5CO]^−^ (Table 1). Based on the comparison with date from the literature, **5** was annotated as brevifolin [26]. Luteolin (**6**) was also identified. The identification of the aglycone was determined based on a series of fragment ions at *m*/*z* 285.041 [Lu-H]^−^, 255.030 [Lu-H-CH_2_O]^−^, 257.042 [Lu-H-CO]^−^, 241.051 [Lu-H-CO_2_]^−^, 227.034 [Lu-H-CH_2_O-CO]^−^, 211.039 [Lu-H-H_2_O-2CO]^−^, together with RDA ions ^1,3^B^−^ at *m*/*z* 133.029, ^1.3^A^−^ at *m*/*z* 151.003 and ^0.4^A^−^ at *m*/*z* 107.012 and comparison with authentic standard (Table 1). All compounds except brevifolin were found previously in *K. africana* [1].

#### 2.1.2. Gallotannins

Herein eight compounds were tentatively identified as gallic acid derivatives. They were esters of gallic acid and polyols, usually hexose. Five gallotannins were identified as monogalloyl hexose (**7** and **8**), digalloyl hexose (**11**), trigalloyl hexose (**12**), and tetragalloyl hexose (**13**). They showed the characteristic fragment ions in their product ion spectra by consecutive elimination of galloyl and gallate moieties. Tetragalloyl hexose (**13**), [M − H]^−^ at *m*/*z* 787.0999 produced a loss of galloyl residue (−152.016 Da) to trigalloyl hexose (at *m*/*z* 635.0884), and fragment ions at *m*/*z* 617.080 [M − H-GA]^−^, 465.067 [M − H-2galloyl-H_2_O]^−^, 313.057 [M − H-3galloyl-H_2_O]^−^ (Table 1, Figure 1). Fragment ions at *m*/*z* 271.0466, 169.0145 and 125.0250 were observed as cross-ring fragment ions of a hexose molecule, and deprotonated and decarboxylated ions of the gallic acid moieties, respectively [27]. Compound **9** [M − H]^−^ at *m*/*z* 169.0142 was identified as gallic acid (GA), based on fragment ions at *m*/*z* 125.023 [GA-H-CO_2_]^−^ and 107.012 [GA-H-H_2_O-CO_2_]^−^, and was further confirmed with an authentic standard [28]. Compound **10** differed from **9** by a CH_2_ group, demonstrated similar fragmentation pathway and was ascribed to methyl gallate [29].

However, preliminary phytochemical profiling revealed the presence of tannins in the *K. africana* stem bark, this is the first report on the presence of gallic acid and gallotannins [30]. Gallic acid was previously found in the stem bark of other Bignoniaceae species as *Mansoa alliacea* (Lam.) AHGentry [31], while tannins were reported for *Stereospermum kunthianum* Cham [32].

#### 2.1.3. Ellagitannins

Ellagitannins (ETs) are different combinations of gallic acid, hexahydroxydiphenic acid and its dilactone ellagic acid with hexose, existing in a wide range of states and forms such as monomers (i.e., ellagic acid glycosides), oligomers, and complex polymers. Herein, thirty-one compounds were dereplicated as ellagic acid derivatives. Compound **36** (C_14_H_6_O_8_), [M − H]^−^ at *m*/*z* 300.9990 gave fragment ions at *m*/*z* 257.007 [M − H-CO_2_]^−^, 229.013 [M − H-CO_2_-CO]^−^, 201.018 [M − H-CO_2_-2CO]^−^, 185.023 [M − H-2CO_2_-2CO]^−^. Thus **36** was identified as ellagic acid and was further confirmed by comparison with authentic standard (Table 1). Compounds **38** and **40** (C_15_H_8_O_8_), ([M − H]^−^ at *m*/*z* 315.0146) differed from **36** by a CH_2_ group and gave an abundant ion at *m*/*z* 299.991 [M − H-CH_3_]^−^. Thus, **38** and **40** were dereplicated as methylellagic acid [29] (Table 1). Similarly, **41**–**43** and **44** were tentatively identified as dimethyl- and trimethylellagic acid, respectively (Figure 2, Table 1). Compound **22** (C_21_H_10_O_13_) differed from ellagic acid with C_7_H_4_O*5* moiety and gave a fragment ion at *m*/*z* 299.99, corresponding to the loss of deprotonated gallic acid. Therefore, **22** was tentatively dereplicated as valoneic acid dilactone (Table 1). Similarly, **29** was related to valoneic acid dilactone methyl ester.

The fragmentation pathways of **30**, **33**, and **35** showed neutral mass losses of hexose (162.053 Da), while **34**, **37**, and **39** demonstrated neutral mass losses of pentose (132.043 Da). Consequently, the aglycone of **30**, **33**, and **34** was registered at *m*/*z* 300.9987, and compounds were ascribed as glycosides of ellagic acid. Compounds **35**/**37** and **39** were identified as glycosides of methylellagic ([Agl-H]^−^ at *m*/*z* 315.0145) and dimethylellagic acid ([Agl-H]^−^ at *m*/*z* at 329.0271), respectively.

MS/MS spectra of compound **14** and **15** revealed fragment ion at *m*/*z* 421.04 [M − H-hex-H_2_O]^−^, *m*/*z* 300.999 corresponding to a deprotonated ellagic acid, and *m*/*z* 257.009 by decarboxylation of the ellagic acid. Hence, compounds were identified as the hexahydroxydiphenic acid (HHDP), esterified with two OH groups of a hexose (HHDP-hexose) (Table 1) [29]. Similarly, compounds **16** and **23**, [M − H]^−^ at *m*/*z* 783.0686 were identified as diHHDP-hexose (pedunculagin) isomers, while **17** was ascribed to castalin/vescalin [29] (Table 1). Compounds **24**/**26** differed from **14**/**15** with a galloyl residue and were related to galloyl-HHDP-hexose (Table 1). Compound **32** [M − H]^−^ at *m*/*z* 600.9896 gave fragment ion at *m*/*z* 300.999 [M − H-2galloyl-2H]^−^ and demonstrated similar to ellagic acid MS/MS spectrum. Thus, **32** was ascribed as gallagic acid dilactone (terminalin), a condensation between ellagic acid and two gallic acids with lactone structures [29].

Two isobars **18** and **19** [M − H]^−^ at *m*/*z* 781.0530 were identified as punicalin isomers (α/β-isomers) based on comparison with literature data [29]. They gave fragment ions at *m*/*z* 600.990 [M − H-hex-H_2_O], 448.979 [M − H-hex-H_2_O-galloyl], and 420.984 [M − H-hex-H_2_O-galloyl-CO], revealing a hexoside and ether bond of the gallic acid residues. Analogically, **20** was tentatively identified as galloylpunicalin [29].

Three isobars **21**, **25**, and **28** shared the same [M − 2H]^2−^ at *m*/*z* 541.0260, corresponding to the doubly-charged punicalagin isomers ions, respectively [28]. Previously, Falode et al., 2017 found that the content of ellagic acid reach up to 4.01 ± 0.01 mg/g in the *K. africana* leaves extract [16]. However, this is the first report on the composition of ellagitannins in *K. africana* stem bark.

#### 2.1.4. Phenylethanoid Glycosides and Iridoids

Three phenylethanoid glycosides (**45**, **47**, and **48**), and two iridoid glycosides (**46** and **49**) were annotated in the studied *K. africana* stem bark extract. The MS/MS spectra of compounds **47** and **48** revealed similar fragmentation patterns with ions at *m*/*z* 461.167 [M − H-caffeoyl]^−^, 315.108 [M − H-caffeoyl-dHex]^−^, and 153.054 [M − H-caffeoyl-dHex-Hex]^−^, together with fragment ions of caffeic acid at *m*/*z* 179.033 [CA-H]^−^, 161.023 [CA-H-H_2_O]^−^, 135.043 [CA-H-CO_2_]^−^. Based on the comparison of MS/MS spectra and retention times to literature data, compounds **47**/**48** were related to verbascoside/acteoside (Table 1) [33]. Analogically, but with an additional glycosyl residue, compound **45** was annotated as echinacoside [14]. The phenylethanoid glycosides were previously found in *K. africana* fruits.

Compound **46** ([M − H]^−^ at *m*/*z* 523.1457) gave fragment ions at 363.094 [M − H-hex]^−^, 343.083 [M − H-hex-H_2_O]^−^, as well as fragments at *m*/*z* 179.033, 161.023, and 135.043, corresponding to the presence of caffeoyl residue. Thus, **46** was dereplicated as verminoside, formerly isolated from *K. africana* fruits [8].

The MS/MS spectrum of compound **49** ([M − H]- at *m*/*z* 481.1715) revealed subsequent losses of hexosyl moiety (−162.053) at *m*/*z* 319.119, CH_3_ group at *m*/*z* 304.095, and fragment ions at *m*/*z* 138.031 corresponding to the hydroxybenzoic acid (Figure 3, Table 1). Hence, **49** was tentatively annotated as methoxybenzoylajugol, previously isolated from the stem bark of *Tabebuia avellanedae*, Bignoniaceae [34].

#### 2.1.5. Naphthoquinones and Anthracene Derivatives

Monoterpenoid naphthoquinones (pinnatal, isopinnatal, kigelinol and isokigelinol) are specific for *K. africana* and other Bignoniaceae species [20]. A key step in the dereplication/annotation of naphto- and anthraquinones was a series of losses of 18 and 28 Da from hydroxyl (OH) and carbonyl (CO) groups on the benzene ring [35], as well as losses of 26 and 44 Da resulted from degradation of the benzene ring (C_2_H_2_) and cleavage of the carboxyl group (CO_2_), respectively [36]. Three isobars **55**, **57**, and **60** shared the same [M − H]^−^ at *m*/*z* 337.1082. They gave fragment ions at *m*/*z* 309.113 [M − H-CO]^−^, 307.097 [M − H-CH_2_O]^−^, 279.066 [M − H-CH_2_O-C_2_H_2_]^−^. Additional indicative ion at *m*/*z* 237.055 (C_15_H_9_O_3_) corresponded to the presence of hydroxymethylanthracene-9,10-dione structure. Consequently, **55**, **57**, and **60** were ascribed to pinnatal and isopinnatal, previously isolated from *K. africana* root and stem bark [6,37] and sterekunthal A, found in the root bark of *Stereospermum kunthianum* (Bignoniaceae) [38]. Compounds **59**/**62** were tentatively identified as kigelinol/isokigelinol based on the deprotonated molecule at *m*/*z* 307.0976 (C_19_H_16_O_4_) and fragments at *m*/*z* 289.087 [M − H-H_2_O], 274.063 [M − H-H_2_O-CH_3_], and 261.091 [M − H-H_2_O-CO], typical for naphto- and anthraquinones. In addition, the significant ion at *m*/*z* 246.068 [M − H-H_2_O-CO-CH_3_] (C_17_H_10_O_2_) corresponded to the presence of condensed structure of anthraquinone and dihydrocyclopentane [39]. Compound **50** ([M − H]^−^ at *m*/*z* 237.004) gave a series of fragments at *m*/*z* 193.013 [M − H-CO_2_]^−^, 149.023 [M − H-2CO_2_]^−^, 121.028 [M − H-2CO_2_-CO]^−^, 93.033 [M − H-3CO_2_-CO]^−^, 77.038 [M − H-3CO_2_-2CO]^−^ and was tentatively annotated as pentahydroxy-1,4-naptoquinone (Table 1). The MS/MS spectrum of **51**/**54** ([M − H at *m*/*z* 285.077) revealed a consecutive loss of two CH_3_ and a two CO groups at *m*/*z* 270.053 [M − H-CH_3_]^−^, 255.029 [M − H-2CH_3_]^−^, 227.034 [M − H-2CH_3_-CO]^−^, and 199.039 [M − H-2CH_3_-2CO]^−^. Additionally, the fragment at *m*/*z* 255.029 (C_14_H_7_O_5_) corresponded to tri-substituted anthraquinone. Hence, **51**/**54** were related to dimethoxy-hydroxyanthraquinone (Table 1). Similarly, **52**, **56**, **58**/**63**, and **61** were tentatively ascribed to dehydroanthraquinones substituted with a variety number of methoxy and hydroxy groups (Table 1). The MS/MS spectrum of **53** gave a base peak at *m*/*z* 331.119, resulting from the loss of hexose (−162.053 Da) and successive losses of four CH_3_ groups. Therefore, **53** were annotated as hydroxy-tetramethoxy-dihydroanthraquinone-*O*-hexoside (Figure 4, Table 1).

All compounds except pinnatal, isopinnatal, kigelinol, and isokigelinol are reported for the first time in *K. africana*. As indicated by the UHPLC analysis, the studied profile of the *K. africana* extract was dominated by the following compounds: trimetylellagic acid (**44**), dimethylellagic acid isomers (**41** and **42**) (8.82 and 8.72%), ellagic acid (**36**) (6.27%), hydroxy-tetramethoxy-dihydroanthraquinone-*O*-hexoside (**53**) (6.30%), punicalgin α/β/γ (**28**) (4.42%), and methylellagic acid isomers (**38** and **40**) (4.22 and 4.20%).

### 2.2. In Vitro Cytotoxicity of K. africana Stem Bark Extract

The anticancer activity of the *K. africana* stem bark extract was evaluated in a panel of malignant human cell lines of different origin in the concentration range of 25–400 µg/mL. The antiproliferative effects of the extract were monitored after 72 h exposure to five serial concentrations using a standard MTT-based colorimetric assay for assessing cell viability. The obtained data were fitted to “concentration–effect” curves by means of non-linear regression (GraphPad Prism 8.0 software) and the corresponding equi-effective concentrations (IC_50_) were calculated. In all tumor models, we established a pronounced inhibition of cell growth in a mostly dose-dependent manner (Figure 5, Figure 6 and Figure 7), with IC_50_ values ranging near and well below (4–30 µg/mL) the lowest treatment concentration of 25 µg/mL (Table 2). Furthermore, in all malignant models was observed a marked tumor selectivity in the cytostatic effects of the *K. africana* extract, as indicated by the calculated selectivity indices (SI, the ratio of the IC_50_ for the normal HEK-293 cells to the IC_50_ for the correspondent malignant cell line), ranging from about 8 to over 50. A further analysis of the experimental results revealed some variations in EKA performance depending on the origin and type of the screened tumor models.

In the cytotoxicity screenings, the two breast cancer models (TNBC MDA-MB-231 cells and hormone-responsive MCF-7 cell line) showed the lowest chemosensitivity to the *Kigelia* extract with almost identical IC_50_ values of about 30 µg/mL and a selectivity index of about 8 (Table 2). In the tested concentration range, both cell lines exhibited very similar cytotoxicity profiles, displaying significant deviation only at the exposure level of 200 µg/mL. At this concentration, the growth-inhibitory effect of the EKA was about seven times stronger against the hormone-sensitive MCF-7 tumor model, as compared to the more resilient TNBC subtype.

In the other pair of epithelial malignancies (T-24 and CAL-29 urothelial carcinomas), comparing cell viability profiles revealed a more defined trend in the mean potency of the tested extract. Accordingly, the anticancer activity of EKA is nearly twice higher against the T-24 tumor model at all but the highest treatment concentrations (25 µg/mL, 50 µg/mL, 100 µg/mL and 200 µg/mL). As a result, more than a two-fold difference was observed in the estimated IC_50_ values (10.2 µg/mL for T-24 and 24.8 for CAL-29 cells, respectively) and the correspondent SIs (23.3 for the T-24 cells and 9.6 for the CAL-29 model). The lower chemosensitivity displayed by the CAL-29 carcinoma model is most likely due to its high expression of the multidrug resistance (MDR1) transporter, which is known to facilitate the cellular efflux of various xenobiotics, including phytochemicals. In addition, CAL-29 cells originate from a patient with primary lesion of a fatally invasive and much more aggressive metastatic transitional cell carcinoma of the bladder (grade IV, stage T2) [40].

As shown in Figure 7 and Table 2, the antineoplastic activity of the EKA was most pronounced in the two models of cutaneous T-cell lymphoma (CTCL), namely Mycosis fungoides (MJ cell line) and Sézary syndrome (HUT-78 cells), exhibiting IC_50_ values of 11.50 µg/mL and 4.63 µg/mL, respectively. Accordingly, the estimated selectivity indices for the lymphoma models are of the order of tens (51.6 for the HUT-78 and 20.6 for the MJ cell line). The HUT-78 T-lymphocytes demonstrated a nearly twice higher susceptibility, which is particularly evident at the two lowest treatment concentrations (25 and 50 µg/mL).

The primary data from the conducted cytotoxicity screenings testify to the extremely high potential of the *K. africana* extract as an antineoplastic remedy of natural origin. Calculated IC_50_ values for all tumor models ranged in the low µg/mL range, qualifying it as a highly potent plant extract, matching or far exceeding the in vitro activity of some of the most toxic plant sources, including *Catharanthus roseus*, *Annona muricata*, and *Curcuma longa* [41,42,43]. Even more valuable in practical terms, the studied extract showed explicit selectivity in its cytotoxic action towards all malignant cell types, with extremely favorable selectivity indices invariably higher than 7–8 (a SI of at least 2 is considered as merely favorable). Moreover, the observed lack of toxicity towards healthy HEK-293 cells is well in line with the results of our previous study [23], evaluating the in vivo effects of the same extract on Lewis lung carcinoma (LLC) bearing mice at far greater exposure levels. The established cytotoxicity profile of the *K. africana* extract, highly biased toward malignantly transformed but not normal cells, suggests specific modulation of defined molecular tumor targets, which will be the subject of future investigation, aiming to further elucidate the relationship between the phytochemical composition and antitumor activity of the extract.

Our findings support previous studies of the anticancer activity of the plant towards other tumor cell lines. Arkhipov et al., 2014 [5] reported the significant antiproliferative activity of *K. africana* fruit extracts against Jeg-3 choriocarcinoma cells. The methanol and water extracts displayed the strongest cytostatic activity, inhibiting Jeg-3 growth to 42% and 46% compared to the untreated control group, respectively [5]. Cytotoxic activity for *K. africana* fruit extracts has also been reported against a melanoma and two breast cancer cell lines. The same study also identified several metabolites, including demethylkigelin, kigelin, ferulic acid, and 2-(1-hydroxyethyl)-naphtho[2,3-b] furan-4,9-dione as inhibiting cancer cell proliferation. The compound 2-(1-hydroxyethyl)-naphtho[2,3-b] furan-4,9-dione was reported to be a particularly strong anticancer agent [15]. Crude dichloromethane extracts of stem bark and fruit showed cytotoxic activity in vitro against cultured melanoma and other cancer cell lines (G361, StML11a, C32, ACHN, Colo 668, CHO) using the sulphorhodamine B assay. Thin layer chromatography examination of the most active fractions of both stem bark and fruits showed the presence of the same major components which were found to be norviburtinal and beta-sitosterol. Norviburtinal was found to be the most active compound but had little selectivity for melanoma cell lines whilst isopinnatal also showed some cytotoxic activity [44]. Recently, the anticancer activity of 1,8-dihydroanthraquinone derivatives in in vitro and in vivo breast cancer models were comprehensively analyzed. Mechanistic studies have established that these compounds exert their anti-breast cancer activities through a wide array of molecular targets and mechanisms, including the modulation of angiogenesis, apoptotic pathways, autophagy, synthesis and repair genes expression, damage response, cell cycle regulators, EMT markers, epigenetic mechanisms, heat shock response, inflammation, metastasis-related markers, oxidative status, miRNA, protein synthesis, proliferation, or stem-like markers [45]. Therefore, the annotated naphthoquinones and anthracene derivatives (**50**–**63**) are likely to be at least partly accountable for the established anticancer activity of the studied extract. Similarly to anthracyclines, naphthoquinones feature a conjugated cyclic dione structure that enables them to serve as electron donors and acceptors and produce extremely toxic hydroxyl radicals leading to lipid peroxidation and ferroptosis in breast, prostate and bladder carcinomas, among others [46,47]. Tricyclic anthracene derivatives, on the other hand, bear a planar polycyclic structure and can readily intercalate into the DNA matrix, thus interfering with topoisomerase II activity during DNA replication [48]. Furthermore, along with other chemically related derivatives, anthracenes and naphthoquinones have been recognized as typical substrates of the MDR1 efflux pump, which would explain the lower activity of the extract in the MDR1 positive CAL-29 bladder carcinoma model [49].

However, the dominating compounds in the studied *K. africana* extract were trimethylellagic acid (**44**), other ellagic acid derivatives (**36**, **40**, **41**, **42**), and ellagitannins (**28**). Wardana et al., 2022 [50] reported the anticancer activity of 3,4,3′-tri-O-methylellagic acid isolated from *Syzygium polycephalum*. The in vitro evaluation showcased the inhibition activity of compound towards the T47D and HeLa cell lines with EC50 values of 55.35 ± 6.28 μg mL^−1^ and 12.57 ± 2.22 μg mL^−1^, respectively. Moreover, the in silico evaluation aimed to elucidate the interaction of tri-*O*-methylellagic acid with enzymes responsible for cancer regulation at the molecular level by targeting the hindrance of cyclin-dependent kinase 9 (CDK9) and sirtuin 1 (SIRT1) enzymes [50]. Numerous in vitro and in vivo studies in different cancer cell lines revealed the antiproliferative properties of ellagic acid [15,51,52,53,54,55]. The antiproliferative action of ellagic acid could be mediated for its ability to directly inhibit the DNA binding of certain carcinogens, including nitrosamines [56] and polycyclic aromatic hydrocarbons [57]. In addition, de Molina et al., 2014 [58] reported that 4,4′-di-*O*-methylellagic acid was the most effective compound in the inhibition of colon cancer cell proliferation among other chemically related compounds. 4,4′-di-*O*-methylellagic acid was highly active against colon cancer cells resistant to the chemotherapeutic agent 5-fluoracil, whereas no effect was observed in non-malignant colon cells [58].

## 3. Materials and Methods

### 3.1. Plant Material

*K. africana* plant material (stem bark) was collected at Durban, South Africa in December 2018 and identified by one of the authors (N.S.) A voucher specimen (1/30.01.2025) was deposited at the Bews Herbarium (NU), School of Life Sciences, of University of KwaZulu-Natal, Durban, South Africa. The plant material was dried at room temperature.

### 3.2. Sample Extraction

Air-dried grounded stem bark (100 g) was extracted twice with 80% MeOH (1:20 *w*/*v*) by sonication (80 kHz, ultra-sound bath Biobase UC-20C, Jinan, China) for 15 min at room temperature. The extracts were concentrated in vacuo and subsequently lyophilized (lyophilizer Biobase BK-FD10P, Jinan, China) to yield crude extracts of 10.20 g.

### 3.3. Chemicals

Acetonitrile (hypergrade for LC–MS), formic acid (for LC–MS), and methanol (an-alytical grade) were purchased from Chromasolv (Sofia, Bulgaria). The reference standards used for compound identification were obtained from Extrasynthese (Genay, France) for gallic, gentisic, *p*-hydroxybenzoic acids, and luteolin. Ellagic and caffeic acids were supplied from Phytolab (Vestenbergsgreuth, Bavaria, Germany). MTT was provided from BLD Pharmatech GmbH (Kaiserslautern, Germany).

### 3.4. UHPLC-HRMS

The UHPLC-HRMS analyses were carried out on a Q Exactive Plus mass spectrometer (ThermoFisher Scientific, Inc., Waltham, MA, USA) equipped with a heated electrospray ionization (HESI-II) probe (ThermoScientific). The equipment was operated in negative and positive ion modes within the m/z range from 100 to 1000. The mass spectrometer parameters were as follows: spray voltage 3.5 kV (+) and 2.5 kV (−); sheath gas flow rate 38; auxiliary gas flow rate 12; spare gas flow rate 0; capillary temperature 320 °C; probe heater temperature 320 °C; S-lens RF level 50; scan mode: full MS (resolution 70,000), and MS/MS (17,500). The chromatographic separation was achieved on a reversed phase column Kromasil EternityXT C18 (1.8 µm, 2.1 × 100 mm) at 40 °C. The UHPLC analyses were run with a mobile phase consisting of 0.1% formic acid in water (A) and 0.1% formic acid in acetonitrile (B). The run time was 33 min. The flow rate was 0.3 mL/min. The gradient elution program was used as follows: 0–1 min, 0–5% B; 1–20 min, 5–30% B; 20–25 min, 30–50% B; 25–30 min, 50–70% B; 30–33 min, 70–95%; 33–34 min 95–5% B. Equilibration time was 4 min [25]. Data were processed by Xcalibur 4.2 (ThermoScientific, Waltham, MA, USA) instrument control/data handling software.

### 3.5. MTT Cell Viability Assay

The antiproliferative activity of the *K. africana* stem bark extract was assessed against normal embryonic kidney cells (HEK-293), as well as malignant human cell lines of different origin (hormone-responsive breast carcinoma cell line, MCF-7; triple negative breast carcinoma cells, MDA-MB-231; urothelial bladder carcinoma cell lines, T-24, CAL-29; cutaneous T-cell lymphoma cell lines, HUT-78, MJ). The antineoplastic activity of the extract was measured using a standard MTT-based colorimetric assay for assessing cell viability. According to protocol, exponential-phased cells were harvested by trypsinization and seeded (100 μL/well) in 96-well plates at the appropriate density of at least 1 × 10^5^ cells/mL. After a 24 h incubation, cells were treated with serial dilutions of the tested extract in the concentration range of 25–400 µg/mL. Following a 72 h exposure, filter sterilized MTT substrate solution (5 mg/mL in PBS) was added to each well of the culture plate. A further 2–4 h incubation allowed the reduction of the yellow MTT reagent into purple formazan crystals in metabolically active viable cells, which were dissolved in isopropyl alcohol solution containing 5% formic acid prior to absorbance measurement at 550 nm. Collected absorbance values were blanked against MTT and isopropanol solution and normalized to the mean value of untreated control (100% cell viability).

### 3.6. Statistical Analysis

The experiments for the evaluation of antiproliferative activity were performed in triplicate and the results were presented as mean and standard deviation. The obtained data were fitted to “concentration-effect” curves by means of non-linear regression (GraphPad Prism 8.0 software) and the corresponding equi-effective concentrations (IC_50_) were calculated. Statistical analysis of the data was performed via a one-way ANOVA (* *p* values ≤ 0.01 are considered statistically significant. 

## 4. Conclusions

The study presents a comprehensive metabolite profiling of the *K. africana* stem bark extract by means of ultra-high-performance liquid chromatography—Orbitrap high-resolution mass spectrometry. A total of 63 secondary metabolites, noticeably including phenolic acids, gallo- and ellagitannins, iridoids, naphthoquinone, and anthraquinones, were dereplicated/annotated. Herein, the majority of annotated gallo- and ellagitannins were reported for the first time in *K. africana*. In all of the tested in vitro models, the studied stem bark extract showed substantial concentration dependent antineoplastic activity, while exhibiting marked tumor selectivity with SIs ranging between 8 and 51. The strongest inhibitory effect of the extract was found on the growth of non-invasive urinary bladder T-24 cancer cells and CTCL-derived malignant cells with estimated IC_50_ concentrations of less than 12 µg/mL, while the breast carcinomas and the invasive bladder carcinoma cell line were slightly less responsive. The isolation of the main compounds and their pharmacological investigation as potential candidate with antiproliferative activity may be a subject of further complementary studies.

## Figures and Tables

**Figure 1 molecules-30-01388-f001:**
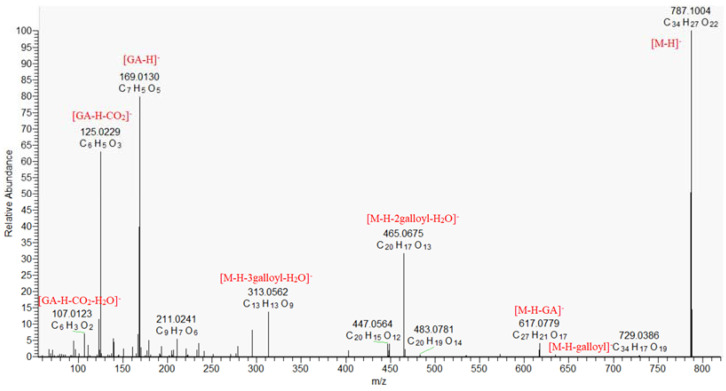
MS/MS spectrum of tetragalloyl hexose (**13**).

**Figure 2 molecules-30-01388-f002:**
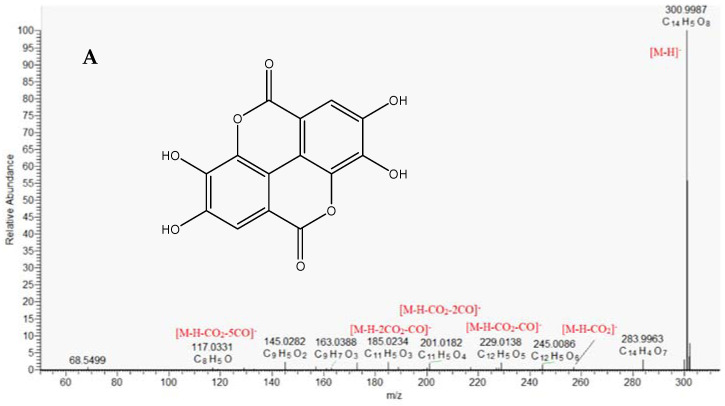
MS/MS spectrum of ellagic acid (**36**) (**A**) and trimethylellagic acid (**44**) (**B**).

**Figure 3 molecules-30-01388-f003:**
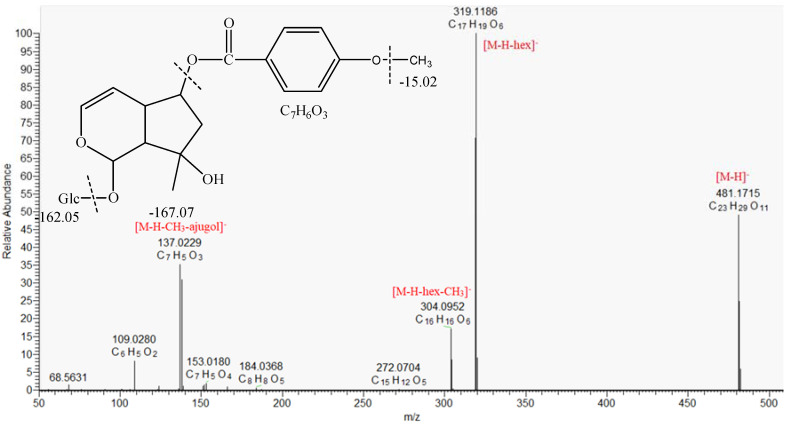
MS/MS spectrum of methoxybenzoylajugol (**49**).

**Figure 4 molecules-30-01388-f004:**
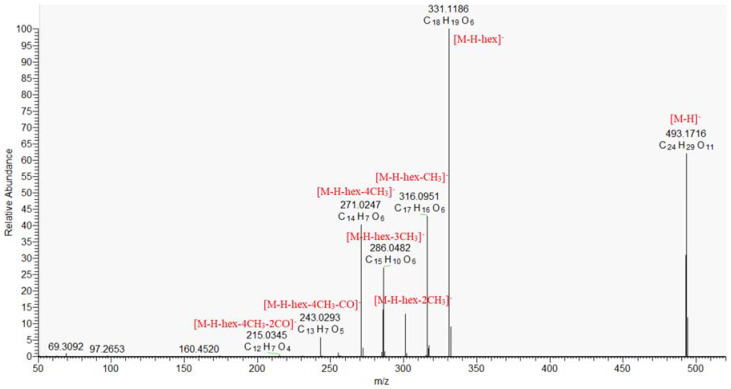
MS/MS spectrum of hydroxy-tetramethoxy-dihydroanthraquinone-*O*-hexoside (**53**).

**Figure 5 molecules-30-01388-f005:**
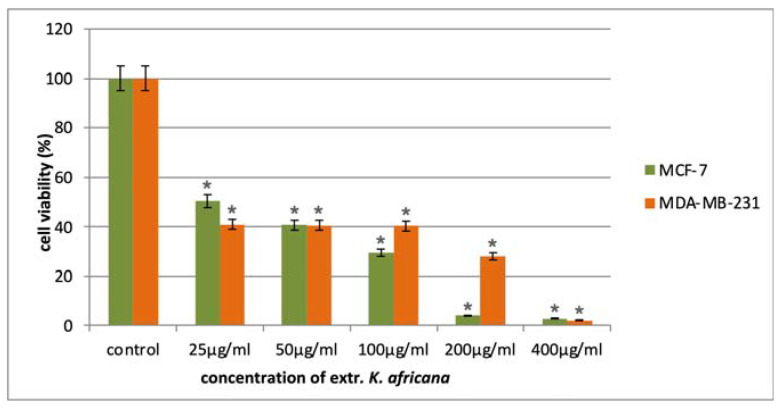
In vitro cytotoxicity of the *K. africana* extract in the tested concentration range against breast carcinoma cell lines (hormone-responsive MCF-7 breast cancer and triple negative MDA-MB-231 breast cancer). All experiments were run in triplicate and data are expressed as mean ± SD. Statistical analysis of the data was performed via one-way ANOVA (* *p*-values ≤ 0.01 are considered statistically significant).

**Figure 6 molecules-30-01388-f006:**
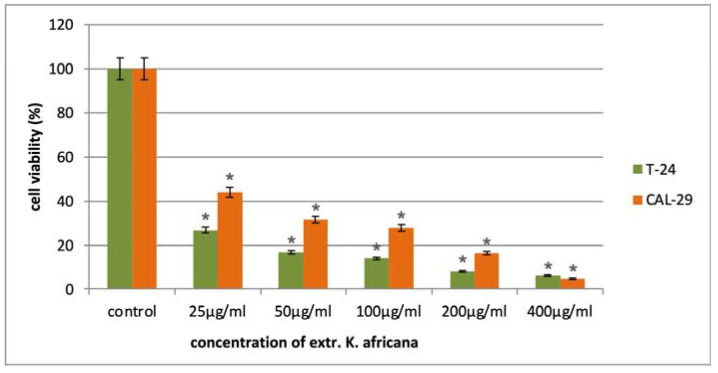
In vitro cytotoxicity of the *K. africana* extract in the tested concentration range against urothelial bladder carcinoma cell lines T-24 and CAL-29. All experiments were run in triplicate and data are expressed as mean ± SD. Statistical analysis of the data was performed via one-way ANOVA (* *p*-values ≤ 0.01 are considered statistically significant).

**Figure 7 molecules-30-01388-f007:**
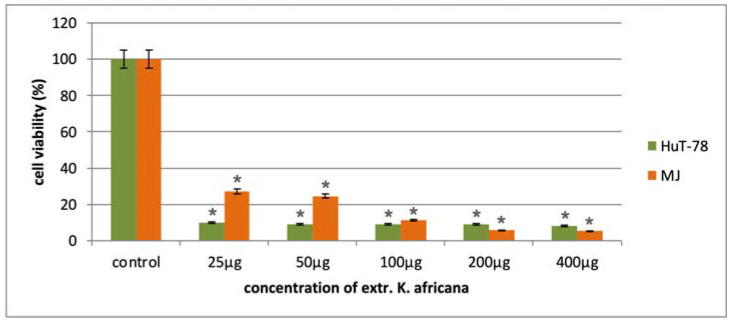
In vitro cytotoxicity of the *K. africana* extract in the tested concentration range against cutaneous T-cell lymphoma cell lines HUT-78 and MJ. All experiments were run in triplicate and data are expressed as mean ± SD. Statistical analysis of the data was performed via one-way ANOVA (* *p*-values ≤ 0.01 are considered statistically significant).

**Table 1 molecules-30-01388-t001:** Secondary metabolites in *Kigelia africana* stem bark methanol-aqueous extract assayed by UHPLC-HRMS.

No	Identified/Tentatively Annotated Compound	Molecular Formula	Exact Mass [M − H]	Fragmentation Pattern in (-) ESI-MS/MS	t_R_ (min)	Δ ppm	Level of Confidence
**Phenolic acids, flavonoids and derivatives**							
**1.**	gentisic acid ^a^	C_7_H_6_O_4_	153.0193	153.0184 (14.5), 109.0279 (100), 81.0328 (1.6)	3.26	−8.574	1
**2.**	dihydroxybenzoic acid *O*-pentoside	C_12_H_14_O_8_	285.0616	285.0615 (92.9), 153.0180 (42.6), 152.0101 (65.7), 123.0070 (0.7), 108.0200 (100), 85.0280 (0.6)	4.18	−0.458	2
**3.**	*p*-hydroxybenzoic acid ^a^	C_7_H_6_O_3_	137.0244	137.0232 (100), 119.0121 (0.9), 108.0201 (6.1), 93.0330 (22.6), 81.0329 (2.5)	4.66	−8.738	1
**4.**	caffeic acid ^a^	C_9_H_8_O_4_	179.0350	179.0343 (19.9), 135.0437 (100), 161.1659 (1.1), 107.0486 (1.3)	6.73	−3.865	1
**5.**	brevifolin	C_12_H_8_O_6_	247.0248	247.0242 (100), 219.0292 (2.9), 201.0181 (0.5), 191.0339 (9.7), 173.0230 (3.1), 163.0389 (2.1), 145.0280 (3.7), 135.0435 (1.1), 119.0488 (1.4), 107.0485 (0.2)	9.02	−2.393	2
**6.**	luteolin ^a^	C_15_H_10_O_6_	285.0405	285.0404 (100), 151.0023 (6.3), 133.0282 (21.4), 107.0122 (5.7)	16.62	−0.320	1
**Gallotannins**							
**7.**	galloyl hexose isomer I	C_13_H_16_O_10_	331.0671	331.0670 (100), 271.0670 (1.0), 241.0361 (1.2), 311.0242 (14.3), 169.0129 (40.3), 151.0024 (13.9), 125.0229 (13.8), 107.0123 (4.5)	1.27	−0.332	2
**8.**	galloyl hexose isomer II	C_13_H_16_O_10_	331.0671	331.0670 (16.3), 271.0458 (100), 241.0343 (1.4), 211.0240 (30.2), 169.0130 (24.9), 125.0229 (19.3)	1.38	−0.151	2
**9.**	gallic acid ^a^	C_7_H_6_O_5_	169.0142	169.0134 (35.79), 125.0229 (100), 123.0071 (0.94), 107.0122 (1.1), 97.0279 (4.11)	1.68	−5.186	1
**10.**	methyl gallate	C_8_H_8_O_5_	183.0299	183.0292 (100), 168.0052 (13.1), 140.0101 (10.9), 127.0020 (2.5), 111.0072 (6.6)	5.35	−3.751	2
**11.**	digalloyl hexose	C_20_H_20_O_14_	483.0780	483.0786 (100), 331.0675 (3.0), 313.0569 (16.7), 271.0462 (52.6), 211.0220 (14.4), 169.0133 (48.6), 125.0229 (36.4)	5.81	1.266	2
**12.**	trigalloyl hexose	C_27_H_24_O_18_	635.0879	635.0895 (90.0), 483.0808 (0.6), 465.0673 (100), 447.0559 (0.5), 313.0565 (47.6), 295.0457 (5.2), 241.0346 (2.2), 211.0240 (8.6), 193.0130 (4.4), 169.0129 (73.5), 125.0229 (60.9), 107.0122 (9.4)	8.22	0.776	2
**13.**	tetragalloyl hexose	C_34_H_28_O_22_	787.0999	787.1002 (100), 635.0847 (0.7), 617.0803 (6.0), 465.0673 (29.8), 447.0556 (2.7), 403.0667 (0.9), 313.0559 (12.5), 300.9977 (0.5), 271.0448 (0.8), 295.0461 (8.5), 251.0532 (0.5), 211.0234 (5.9), 193.0129 (2.1), 169.0130 (75.6), 125.0229 (65.6)	10.92	0.349	2
**Ellagitannins**							
**14.**	HHDP-hexose isomer I	C_20_H_18_O_14_	481.0624	481.0623 (100), 300.9989 (66.3), 275.0197 (36.9), 257.0085 (6.1), 229.0136 (10.6), 201.0185 (6.1), 173.0233 (2.8), 145.0289 (2.5), 123.0074 (1.6)	0.83	−0.246	2
**15.**	HHDP-hexose isomer II	C_20_H_18_O_14_	481.0624	481.0624 (100), 421.0404 (0.4), 300.9988 (76.5), 275.0196 (40.3), 257.0087 (6.4), 229.0137 (11.7), 201.0184 (5.0), 173.0233 (3.2), 145.0278 (1.9), 123.0069 (1.8)	1.15	0.128	2
**16.**	pedunculagin isomer I	C_34_H_24_O_22_	783.0686	783.0694 (68.4), 481.0621 (0.9), 300.9988 (100), 275.0196 (40.5), 249.0402 (5.4), 229.0136 (14.6), 145.0285 (3.5), 123.0068 (0.7)	2.94	0.964	2
**17.**	castalin/vescalin	C_27_H_20_O_18_	631.05768	631.0579 (100), 450.9943 (88.2), 432.9821 (4.0), 425.0160 (6.3), 407.0040 (1.1), 379.0106 (2.3), 351.0152 (3.2), 323.0195 (2.8), 295.0253 (2.3), 279.0285 (0.6), 267.0292 (2.0)	3.78	0.370	2
**18.**	punicalin α/β	C_34_H_22_O_22_	781.0530	781.0535 (100), 448.9783 (48.3), 420.9829 (5.4), 392.9886 (52.1), 298.9839 (0.8), 600.9926 (0.7), 364.9933 (7.3), 336.9989 (19.7), 309.040 (3.4), 300.9981 (0.6), 265.0138 (3.1), 237.0179 (2.9), 123.0071 (0.6)	3.99	0.685	2
**19.**	punicalin α/β	C_34_H_22_O_22_	781.0530	781.0538 (100), 600.9902 (0.8), 448.9786 (53.9), 420.9838 (4.5), 392.9891 (56.9), 364.9934 (5.1), 336.9993 (19.2), 321.0037 (2.5), 281.0090 (3.0), 237.0186 (2.6), 209.0237 (1.3), 166.9978 (0.8), 123.0069 (1.2), 299.9911 (0.9)	4.16	1.069	2
**20.**	2-*O*-galloylpunicalin	C_41_H_26_O_26_	933.0640	933.0644 (91.2), 889.3828 (0.3), 781.0530 (44.4), 721.0327 (2.3), 600.9899 (13.5), 450.9941 (100), 425.0152 (10.4), 300.9986 (32.3), 249.0177 (0.5), 229.0132 (3.2), 145.0280 (6.6)	4.18	0.468	2
**21.**	punicalgin α/β/γ	C_24_H_14_O_15_	541.0260	541.0261 (100), 450.9945 (3.4), 300.9987 (40.9), 275.0195 (17.0), 229.0139 (7.2), 173.0227 (2.6), 145.0282 (2.3)	4.19	0.124	2
**22.**	valoneic acid dilactone	C_21_H_10_O_13_	469.0048	469.0045 (48.8), 425.0150 (100), 407.0042 (15.1), 379.0101 (5.6), 351.0135 (2.6), 335.0205 (2.4), 307.0253 (1.5), 300.9968 (3.7), 299.9908 (15.9), 251.0348 (1.7), 223.0401 (0.6), 207.0450 (0.9), 172.0160 (0.6), 145.0281 (0.5)	4.25	−0.860	3
**23.**	pedunculagin isomer I	C_34_H_24_O_22_	783.0686	783.0696 (69.6), 481.0630 (0.8), 300.9989 (100), 275.0197 (51.7), 249.0404 (7.8), 229.0138 (11.8), 145.0285 (2.9), 123.0075 (0.9)	4.56	1.270	2
**24.**	galloyl-HHDP-hexose	C_27_H_22_O_18_	633.0733	633.0738 (100), 300.9988 (81.0), 275.0197 (55.6), 257.0090 (8.2), 229.0138 (13.7), 201.0184 (6.8), 169.0132 (4.8), 125.0228 (3.9)	4.66	0.716	2
**25.**	punicalgin α/β/γ	C_24_H_14_O_15_	541.0260	541.0259 (100), 300.9987 (41.99), 275.0197 (21.3), 229.0133 (5.4), 173.0231 (2.8), 125.0229 (2.1)	5.64	−0.098	2
**26.**	galloyl-HHDP-hexose isomer	C_27_H_22_O_18_	633.07333	633.0737 (100), 300.9988 (75.0), 275.0196 (48.4), 257.0080 (7.7), 229.0136 (12.5), 201.0183 (5.7), 169.0127 (4.5), 125.0229 (4.6), 107.0124 (1.6)	6.13	0.621	2
**27.**	tellimagrandin I	C_34_H_26_O_22_	785.0843	785.0850 (95.8), 483.0754 (0.8), 313.0577 (1.6), 300.9987 (100), 275.0496 (35.5), 249.0401 (30.6), 229.0134 (10.6), 169.0129(17.4), 125.0230 (16.6), 107.0120 (2.6)	7.37	0.923	2
**28.**	punicalgin α/β/γ	C_24_H_14_O_15_	541.0260	541.0259 (100), 450.9940 (43.3), 425.0150 (27.3), 300.9987 (32.9), 229.0137 (10.5), 169.0131 (6.9), 125.0231 (15.3), 173.0233 (6.8)	9.08	−0.098	2
**29.**	methylvaloneic acid dilactone	C_22_H_12_O_13_	483.0205	483.0204 (100), 450.9944 (57.5), 432.09844 (5.2), 407.0046 (4.1), 379.0081 (2.8), 299.9908 (2.4)	9.19	−0.318	2
**30.**	ellagic acid-*O*-hexoside	C_20_H_16_O_13_	463.0518	463.0526 (100), 300.9991 (83.3), 299.9922 (32.1), 107.0343 (2.7)	9.20	1.590	2
**31.**	3,4,8,9,10-pentahydroxy dibenzo[b,d]pyran-6-on	C_13_H_8_O_7_	275.0197	275.0196 (100), 257.0088 (9.6), 247.0242 (1.0), 229.0134 (10.3), 219.0290 (1.4), 203.0339 (3.5), 191.0338 (2.2), 185.0233 (1.9), 173.0233 (2.9), 145.0278 (1.8), 129.0331 (0.3), 101.0382 (0.3)	9.22	−0.385	2
**32.**	gallagic acid dilactone (terminalin)	C_28_H_10_O_16_	600.9896	600.9901 (100), 300.9991 (10.9), 298.9830 (22.6), 270.9883 (14.7), 257.0099 (0.4), 242.9931 (6.5), 229.0154 (1.7), 201.0182 (0.7), 185.0231 (0.6)	10.16	0.737	2
**33.**	ellagic acid-*O*-hexoside isomer	C_20_H_16_O_13_	463.052	463.0525 (100), 373.0215 (16.2), 343.0091 (9.9), 315.0135 (12.9), 300.9991 (1.6), 299.9909 (13.2), 285.0038 (2.1)	10.99	1.461	2
**34.**	ellagic acid-*O*-pentoside	C_19_H_14_O_12_	433.0412	433.0414 (100), 300.9987 (77.2), 299.9911 (42.4), 283.9970 (1.2), 243.9996 (1.9), 185.0230 (1.8)	10.95	0.372	2
**35.**	methylellagic acid-*O*-hexoside	C_21_H_18_O_13_	477.0674	477.0674 (100), 315.0157 (76.9), 314.0067 (7.3), 299.9903 (68.0), 270.9878 (8.7)	11.36	−0.050	2
**36.**	ellagic acid ^a^	C_14_H_6_O_8_	300.9990	300.9991 (100), 245.0664 (1.5), 257.0074 (1.1), 229.0131 (4.1), 201.0184 (2.9), 185.0234 (2.3), 145.0281 (2.8), 117.0333 (0.4)	11.57	0.858	1
**37.**	methylellagic acid-*O*-pentoside	C_20_H_16_O_12_	447.0569	447.0572 (100), 315.0145 (80.1), 314.0072 (4.2), 299.9912 (64.5), 298.9819 (12.8), 270.9869 (19.5)	13.40	0.718	2
**38.**	methylellagic acid	C_15_H_8_O_8_	315.0146	315.0150 (84.6), 299.9912 (100), 270.9866 (2.3), 242.9920 (1.9), 200.0096 (1.7), 300.9955 (11.5)	14.45	2.443	2
**39.**	dimethylellagic acid-O-pentoside	C_21_H_18_O_12_	461.0725	461.0735 (100), 329.0271 (7.9), 328.0230 (54.1), 314.0027 (2.8), 312.9988 (33.9), 297.9762 (30.4), 285.0045 (8.2), 269.9809 (21.9)	14.73	2.041	2
**40.**	methylellagic acid isomer	C_15_H_8_O_8_	315.0146	315.0149 (100), 299.9912 (89.1), 242.9920 (1.7), 200.0092 (2.1), 300.9946 (6.3)	14.78	1.872	2
**41.**	dimethylellagic acid	C_16_H_10_O_8_	329.0303	329.0305 (100), 314.0070 (76.0), 298.9834 (28.8), 285.0030 (6.9), 270.9883 (49.4), 242.9905 (5.4), 214.9963 (1.4), 315.0101 (7.7), 312.9991 (17.8)	18.18	1.609	2
**42.**	dimethylellagic acid isomer I	C_16_H_10_O_8_	329.0303	329.0303 (94.6), 314.0071 (100), 298.9835 (30.6), 270.9986 (54.5), 242.9902 (8.4), 315.0095 (10.4), 312.9980 (3.8)	18.58	1.883	2
**43.**	dimethylellagic acid isomer II	C_16_H_10_O_8_	329.0303	329.0304 (100), 314.0070 (43.7), 315.0094 (3.1), 312.9990 (18.8), 298.9836 (19.9), 285.0052 (8.6), 270.9888 (27.6), 242.9902 (1.7)	19.04	1.974	2
**44.**	trimethylellagic acid	C_17_H_12_O_8_	343.0459	343.0461 (98.2), 328.0227 (100), 312.9994 (48.5), 297.9756 (38.1), 285.0042 (9.9), 269.9805 (29.7), 197.9932 (2.4), 145.9990 (0.7)	22.90	0.232	2
**Phenylethanoid glycosides and iridoids**							
**45.**	echinacoside	C_35_H_46_O_20_	785.2510	785.2520 (66.4), 623.2211 (8.9), 461.1661 (1.5), 179.0337 (1.9), 161.0231 (100), 133.0280 (43.7), 123.0435 (3.2), 135.0436 (12.3)	9.71	1.329	2
**46.**	verminoside	C_24_H_28_O_13_	523.1457	523.1464 (100), 361.0936 (0.9), 343.0828 (0.9), 247.0587 (4.2), 180.0364 (0.9), 179.0337 (18.4), 163.0387 (12.9), 161.0231 (75.3), 135.0437 (24.1), 133.0281 (28.9)	11.70	2.171	2
**47.**	verbascoside/acteoside	C_29_H_36_O_15_	623.1981	623.1995 (72.2), 461.1653 (3.9), 315.1112 (1.7), 179.0338 (2.0), 161.0232 (100), 153.0542 (1.1), 135.0438 (6.7), 133.0280 (35.7)	11.99	2.209	2
**48.**	verbascoside/acteoside	C_29_H_36_O_15_	623.1981	623.1994 (87.8), 461.1672 (11.1), 315.1119 (2.1), 179.0335 (2.7), 161.0233 (100), 153.0548 (0.3), 135.0436 (16.1), 133.0231 (29.9)	13.02	2.016	2
**49.**	methoxybenzoylajugol	C_23_H_30_O_11_	481.1715	481.1724 (50.2), 319.1180 (100), 304.0956 (17.1), 152.0101 (1.1), 138.0309 (34.3), 137.0230 (35.0), 109.0280 (12.4)	16.35	1.715	3
**Naphthoquinones and anthracene derivatives**							
**50.**	pentahydroxy-1,4-naptoquinone	C_10_H_6_O_7_	237.0041	237.0035 (22.3), 193.0132 (100), 149.0229 (91.5), 121.0278 (66.2), 107.0127 (0.6), 93.0329 (2.8), 77.0381 (0.6)	4.36	−2.598	3
**51.**	dimethoxy-hydroxy-dihydroanthraquinone	C_16_H_14_O_5_	285.0768	285.0767 (42.9), 270.0532 (84.5), 255.0295 (100), 227.0343 (57.9), 199.0392 (4.8)	17.66	−0.445	3
**52.**	dihydroxy-methoxyantraquinone	C_15_H_10_O_5_	269.0455	269.0457 (100), 254.0199 (6.4), 253.0142 (31.2), 237.0163 (9.9), 226.0241 (9.0), 185.0225 (4.4), 161.0228 (8.7)	18.01	0.756	3
**53.**	hydroxy-tetramethoxy-dihydroanthraquinone-*O*-hexoside	C_24_H_30_O_11_	493.1715	493.1724 (56.0), 331.1190 (100), 316.0954 (50.7), 301.0721 (13.6), 286.0484 (28.8), 271.0250 (48.0), 257.0435 (0.8), 227.0307 (0.5), 243.0296 (4.9)	18.21	1.795	3
**54.**	dimethoxy-hydroxy-dihydroanthraquinone	C_16_H_14_O_5_	285.0768	285.0768 (24.1), 270.0533 (100), 255.0296 (2.4), 227.0341 (36.9), 199.0365 (2.9), 178.9915 (3.3)	21.26	−0.340	3
**55.**	pinnatal/isopinnatal/sterekunthal A	C_20_H_18_O_5_	337.1082	337.1081 (100), 309.1129 (3.0), 307.0972 (2.6), 294.0889 (0.3), 279.0661 (1.2), 251.0345 (6.6), 237.0551 (29.6), 223.0395 (5.3), 209.0604 (1.8), 195.0432 (0.2), 160.0149 (0.4)	22.52	−0.169	2
**56.**	hydroxy-trimethoxy- dihydroanthraquinone	C_17_H_16_O_5_	299.0925	299.0926 (92.5), 284.0692 (100), 269.0455 (99.8), 254.0221 (40.8), 241.0477 (5.4), 230.0210 (15.0), 226.0266 (67.2), 198.0302 (6.4), 92.9940 (1.7)	22.99	0.311	3
**57.**	pinnatal/isopinnatal/sterekunthal A	C_20_H_18_O_5_	337.1082	337.1081 (100), 321.0780 (0.3), 307.0999 (0.3), 295.2039 (0.3), 267.0670 (1.1), 279.0670 (0.2), 251.0348 (0.4), 237.0551 (36.9), 223.0394 (5.9), 209.0600 (1.2), 198.0022 (0.2), 163.0394 (0.2)	23.14	−0.169	2
**58.**	tetramethoxy-hydroxy-dihydroanthraquinone	C_18_H_20_O_6_	331.1187	331.1190 (100), 316.0953 (95.6), 301.0717 (64.1), 286.0482 (22.6), 271.0248 (76.3), 243.0295 (25.0), 227.0329 (1.6), 215.0324 (3.9), 199.0365 (1.1)	23.21	0.841	3
**59.**	kigelinol/isokigelinol	C_19_H_16_O_4_	307.0976	307.0973 (100), 289.0868 (23.9), 274.0633 (9.8), 261.0920 (0.3), 246.0672 (0.2), 237.1321 (0.2)	23.42	−0.854	2
**60.**	pinnatal/isopinnatal/sterekunthal A	C_20_H_18_O_5_	337.1082	337.1082 (100), 309.1133 (7.4), 307.0974 (5.9), 294.0902 (1.0), 279.0660 (1.2), 251.0349 (11.0), 237.0552 (18.4), 223.0395 (7.9), 209.0600 (2.5), 195.0446 (0.2), 173.0235 (0.4), 160.0150 (0.4)	23.51	−0.169	2
**61.**	dimethoxy-dihydroanthraquinone	C_16_H_14_O_4_	269.0819	269.0819 (56.8), 254.0583 (100), 239.0320 (8.5), 228.9896 (15.3), 211.0394 (65.9), 182.9854 (2.3), 154.9913 (6.9)	23.84	0.029	3
**62.**	kigelinol/isokigelinol	C_19_H_16_O_4_	307.0976	307.0974 (100), 289.0869 (25.0), 274.0634 (9.9), 261.0910 (0.3), 246.0679 (0.4)	24.00	−0.463	2
**63.**	tetramethoxy-hydroxy-dihydroanthraquinone isomer	C_18_H_20_O_6_	331.1187	331.1190 (100), 316.0953 (95.6), 301.0717 (64.1), 286.0482 (22.6), 271.0248 (76.3), 243.0295 (25.0), 227.0329 (1.6), 215.0324 (3.9), 199.0365 (1.1)	24.12	0.841	3

^a^—Compared to a reference standard; Level 1—compounds identified by comparison to reference standard; level 2—putatively annotated compounds, 3—putatively characterized compound classes.

**Table 2 molecules-30-01388-t002:** In vitro cytotoxicity of the *K. africana* extract against a panel of human malignant cell lines of different origin and normal embryonic kidney cells (IC_50_ [μg/mL ± SD).

Cell Line	MCF-7	MDA-MB-231	T-24	CAL-29	HUT-78	MJ	HEK-293
** *K. africana* **	30.3 ± 5.3	29.5 ± 4.8	10.2 ± 1.5	24.8 ± 3.9	4.6 ± 1.6	11.5 ± 2.5	237.5 ± 13.6
**SI ***	7.8	8.0	23.3	9.6	51.6	20.6	

MCF-7: hormone-responsive breast carcinoma cell line; MDA-MB-231: triple negative breast carcinoma cell line; T-24, CAL-29: urothelial bladder carcinoma cell lines; HUT-78, MJ: cutaneous T-cell lymphoma cell lines; HEK-293: normal human embryonic kidney cell line; * SI: selectivity index = IC_50_, normal cell line/IC_50_ malignant cell line.

## Data Availability

The original contributions presented in the study are included in the article; further inquiries can be directed to the corresponding author/s.

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
