# Peer review of "New Insights into the Metabolic Profile and Cytotoxic Activity of Kigelia africana Stem Bark"

_molecules, 2025, doi:10.3390/molecules30061388_

Round 1
Reviewer 1 Report
Comments and Suggestions for Authors
This is an interesting work on plant bark extracts of K. africana and on the potential anticancer applications but needs to be carefully revised according to the following major points before it could be accepted for publication.
line 56: I read "6-hydroxy liteolin," which appears to be a typographical error. I believe the correct term should be "6-hydroxy luteolin," referring to a known flavonoid compound. Please revise the text . In general, please check all compound names throughout the manuscript to ensure they are correctly spelled
Line 74: The reference [https://doi.org/10.1002/bmc.4979] should be cited as a numerical reference. same for [10.1007/BF00117707] at line 373
Line 101: The final sentence of the introduction, "The isolation of the main compounds and their investigation as potential candidate with antiproliferative activity may be a subject of further complementary studies," seems more appropriate as a future perspective rather than as a concluding statement for the introduction.
The authors should introduce the following statement regarding the importance of plants in medicine as living tools for xenobiotic metabolism: "Notably, wild plants, including K. africana, hold significant potential not only for direct therapeutic approaches but also for detoxifying harmful substances (xenobiotics)." cite at least: DOIs 10.3390/jox14040084 and 10.1007/s00580-019-03004-y
Tab 1: check ALL carefully!! for example, the authors refer to tellimagrandin (C27H22O18). Could you please clarify whether this is tellimagrandin I or II? If it is tellimagrandin I, please note that it is a C34 compound, not C27. See pubchem. I recommend carefully checking this entry, as well as ALL other entries in this table, for accuracy.
Figures 1–5: If possible, please change the grey background to white.
section 2.2: 1) it would be useful to include references to bark extracts from other plants, that are currently recommended and/or used as therapeutics, even only in traditional medicine, along with their IC50 values in similar models. A comparison with the literature would provide better context for the reported IC50 values and help the reader assess the potential therapeutic potency of the K. africana extract. Without such comparisons, it is difficult to deternine the significance of the reported IC50 values solely from the numerical entries. 2) It would be beneficial to include a comparison of the K. africana extract's cytotoxicity with healthy cell lines in addition to the cancer cell lines already tested. This would provide more insight into the selectivity and safety of the extract. The results of this comparison should then be incorporated into both the abstract and the conclusions to give a more complete picture of the extract's therapeutic potential.
Comments on the Quality of English LanguageIn general, the English level of the manuscript should be improved. Some sentences could be rephrased for better clarity and flow. typos are also present
Author Response
Reviewer 1
This is an interesting work on plant bark extracts of K. africana and on the potential anticancer applications but needs to be carefully revised according to the following major points before it could be accepted for publication.
Reviewer: line 56: I read "6-hydroxy liteolin," which appears to be a typographical error. I believe the correct term should be "6-hydroxy luteolin," referring to a known flavonoid compound. Please revise the text . In general, please check all compound names throughout the manuscript to ensure they are correctly spelled
Response: Thank you for your note. Corrections have been made.
Reviewer: Line 74: The reference [https://doi.org/10.1002/bmc.4979] should be cited as a numerical reference. same for [10.1007/BF00117707] at line 373
Response: Thank you for the valuable comment. All references have been cited as a numerical reference.
Reviewer: Line 101: The final sentence of the introduction, "The isolation of the main compounds and their investigation as potential candidate with antiproliferative activity may be a subject of further complementary studies," seems more appropriate as a future perspective rather than as a concluding statement for the introduction.
Response: Thank you for the recommendation. The sentence has been removed from the Introduction.
Reviewer: The authors should introduce the following statement regarding the importance of plants in medicine as living tools for xenobiotic metabolism: "Notably, wild plants, including K. africana, hold significant potential not only for direct therapeutic approaches but also for detoxifying harmful substances (xenobiotics)." cite at least: DOIs 10.3390/jox14040084 and 10.1007/s00580-019-03004-y
Response: Thank you for the recommendation. The sentence in question and references have been added in the Introduction section.
Reviewer: Tab 1: check ALL carefully!! for example, the authors refer to tellimagrandin (C27H22O18). Could you please clarify whether this is tellimagrandin I or II? If it is tellimagrandin I, please note that it is a C34 compound, not C27. See pubchem. I recommend carefully checking this entry, as well as ALL other entries in this table, for accuracy.
Response: Thank you for your note. All compounds’ structures have been checked and corrected.
Reviewer: Figures 1–5: If possible, please change the grey background to white.
Response: Thank you for the recommendation. The Figures’ background has been changed to white as requested.
Reviewer : section 2.2: 1) it would be useful to include references to bark extracts from other plants, that are currently recommended and/or used as therapeutics, even only in traditional medicine, along with their IC50 values in similar models. A comparison with the literature would provide better context for the reported IC50 values and help the reader assess the potential therapeutic potency of the K. africana extract. Without such comparisons, it is difficult to deternine the significance of the reported IC50 values solely from the numerical entries. 2) It would be beneficial to include a comparison of the K. africana extract's cytotoxicity with healthy cell lines in addition to the cancer cell lines already tested. This would provide more insight into the selectivity and safety of the extract. The results of this comparison should then be incorporated into both the abstract and the conclusions to give a more complete picture of the extract's therapeutic potential.
Response: Thank you for the valuable recommendation. A review has been made of the existing research literature on the biological activity of the extract of K. africana and relevant articles have been cited. As requested, additional experiments have been performed to evaluate the cytotoxic activity of the extract against normal embryonic kidney cells (HEK-293), which served to assess the presence of tumor selectivity for each malignant cell line.
Reviewer: In general, the English level of the manuscript should be improved. Some sentences could be rephrased for better clarity and flow. typos are also present
Response: Thank you for your note. The English quality of the manuscript has been improved.
Reviewer 2 Report
Comments and Suggestions for Authors
In general, a very interesting article, generally well written, with a clear scientific problem defined; However, there are some methodological aspects that must be clarified, that translate, also, in results/discussion section.
- Abstract:
- K. africana should always be in italic; please verify through the article
- The aim of the article is not explicit in the abstract and it should be - please correct
- Introduction:
- please verify the reference in line 74 (there is no number, but a link) and rewrite the sentence in lines 72-74; scientific language must not be adjectivated supperfully (the term extraordinary regarding a drug is not understandable)
- line 77: instead of application I suggest applications
- rewrite the sentence in lines 77-79, in order to place the numeric references at the end of the sentence
- rewrite the sentence in lines 100-103 because there seems to be part of the results/discussion section and not introduction
- Results and Discussion
- table 1 wul be better perceived if the page is horizontally oriented
- I do not understand the rleevance of the existance of figure 1, taking into consideration the existance of table 1; I suggest the elimination of the figure or its transition to supplementary data
- there should have been a statiscial analysis in the citotoxicity assays; I strongly suggest this to be performed
- table 2 is not necessary; only the IC50 data; please rearrange dat aand discuss it, regarding the IC50 values obtained (how can we classify the extract regarding its citotoxicity)
- a reference regarding a positive control should have been made; please verify
- a refernece regarding the inclusion of the cell lines/types of cancer in study should have been made; please verify
Author Response
Reviewer 2
In general, a very interesting article, generally well written, with a clear scientific problem defined; However, there are some methodological aspects that must be clarified, that translate, also, in results/discussion section.
- Abstract:
Reviewer: K. africana should always be in italic; please verify through the article
Response: Thank you for noticing. Requested corrections have been made.
Reviewer: The aim of the article is not explicit in the abstract and it should be - please correct
Response: Thank you for your recommendation. The aim of the present study has been clarified in the abstract.
- Introduction:
Reviewer - please verify the reference in line 74 (there is no number, but a link) and rewrite the sentence in lines 72-74; scientific language must not be adjectivated supperfully (the term extraordinary regarding a drug is not understandable)
Response: Thank you for the recommendation. The references have been verified. The sentence in lines 72-74 has been modified.
Reviewer: line 77: instead of application I suggest applications
Response: The suggested correction has been made.
Reviewer: rewrite the sentence in lines 77-79, in order to place the numeric references at the end of the sentence
Response: The suggested correction has been made.
Reviewer: rewrite the sentence in lines 100-103 because there seems to be part of the results/discussion section and not introduction
Response: The suggested correction has been made.
- Results and Discussion
Reviewer: table 1 wul be better perceived if the page is horizontally oriented
Response: Thank you for the recommendation. The suggested correction has been made.
Reviewer: I do not understand the rleevance of the existance of figure 1, taking into consideration the existance of table 1; I suggest the elimination of the figure or its transition to supplementary data
Response: Thank you for the recommendation. Figure 1 has been removed.
Reviewer: there should have been a statiscial analysis in the citotoxicity assays; I strongly suggest this to be performed
Response: Thank you the recommendation. Statistical analysis has been provided in the revised version.
Reviewer: table 2 is not necessary; only the IC50 data; please rearrange dat aand discuss it, regarding the IC50 values obtained (how can we classify the extract regarding its citotoxicity)
Response: Thank you for the recommendation. As requested, Table 2 has been simplified.
Reviewer: a reference regarding a positive control should have been made; please verify.
Response: Thank you for your suggestion. A comparative evaluation of the antitumor potential of the studied extract would certainly be beneficial. Our main concern is that the biological activity of potent cytostatics (often active in nano- and low micromolar concentrations) is widely unmatched by that of phytoextracts (where treatment concentrations are measured in µg/mL). Furthermore, the highly heterogeneous origin of the screened cell lines complicates the choice of a single universal reference to be used as a positive control in all malignant models (e.g., for breast carcinomas, an anthracycline or platinum drug would be appropriate, which are irrelevant and unsuitable for cutaneous T-cell lymphoma models).
Reviewer: a refernece regarding the inclusion of the cell lines/types of cancer in study should have been made; please verify
Response: Thank you for your note. The origin of the screened cell lines has been referred to.
Reviewer 3 Report
Comments and Suggestions for Authors
This study tackles two key questions that are linked to each other. First off, the authors are using UHPLC-HRMS to get a complete picture of the metabolic profile of Kigelia africana stem bark extract, with a special focus on identifying secondary metabolites, especially gallo- and ellagitannins. Then, they’re checking out how cytotoxic the extract is against various human cancer cell lines. Basically, they’re trying to fill in the gaps in our understanding of the stem bark’s phytochemical makeup and see if it’s a good source of antiproliferative agents.
What’s new here is that the authors have identified 34 compounds in K. africana stem bark for the first time, including gallotannins, ellagitannins (like pedunculagin isomers and punicalin), and some new anthraquinone derivatives. Most past studies have looked at the fruits, leaves, or root bark, but they’re focusing on the stem bark, which hasn’t been explored much. They’re also connecting the dots between phytochemistry and pharmacology by linking the metabolites they found (like trimethylellagic acid) to the cytotoxic effects they observed. This helps address the lack of detailed info on how the metabolites in this plant part relate to their antiproliferative activity.
By identifying 63 metabolites, including 19 new ellagitannins and 7 anthraquinones, this work is expanding what we know about the phytochemicals in K. africana. The cytotoxic evaluation against six cancer models (like TNBC and CTCL) is also new, since most previous studies tested fruit extracts or isolated compounds (like lapachol). The IC50 values they found for the stem bark extract (4–30 µg/mL) are pretty strong compared to what’s been reported before (like 42–55 µg/mL for fruit extracts against Jeg-3 cells). By combining the UHPLC-HRMS data with the bioactivity results, they’re getting a comprehensive view of the therapeutic potential of the stem bark.
Overall, it is a good manuscript. However, the current version is not suitable for acceptance for publication. The following comments or questions may help improve the manuscript.
- In metabolite quantification experiments, relative abundances (e.g., 8.82% abundance of dimethylellagic acid isomers) lack context.
- The mechanisms linking specific metabolites (e.g., hydroxytetramethoxydihydroanthraquinone-O-hexoside) to cytotoxicity remain speculative and need to be discussed and strengthened.
- Details in the figures and tables need to be strengthened.
- **Table 1**: The “Confidence” column lacks clarity—define levels (e.g., Level 1: identified via standard) in the caption. Include retention time variability for isomers (e.g., punicalin α/β).
- **Figures 2–5**: MS/MS spectra are illustrative but lack high-resolution labels (e.g., *m/z* values). Annotate key fragments directly on spectra for readability.
- **Figure 1**: The TIC should align peak numbers with Table 1 entries (e.g., peak 44 = trimethylellagic acid).
- **Data quality**: High-resolution MS data (ppm errors <5) and dose-response curves (R2 values) support robustness.
- Some text formats require attention, such as IC50 in line 438.
- The brand and manufacturer information of the MTT reagent needs to be indicated.
- “5. Conclusions” shold be “4. Conclusions”.
Author Response
Reviewer 3
This study tackles two key questions that are linked to each other. First off, the authors are using UHPLC-HRMS to get a complete picture of the metabolic profile of Kigelia africana stem bark extract, with a special focus on identifying secondary metabolites, especially gallo- and ellagitannins. Then, they’re checking out how cytotoxic the extract is against various human cancer cell lines. Basically, they’re trying to fill in the gaps in our understanding of the stem bark’s phytochemical makeup and see if it’s a good source of antiproliferative agents.
What’s new here is that the authors have identified 34 compounds in K. africana stem bark for the first time, including gallotannins, ellagitannins (like pedunculagin isomers and punicalin), and some new anthraquinone derivatives. Most past studies have looked at the fruits, leaves, or root bark, but they’re focusing on the stem bark, which hasn’t been explored much. They’re also connecting the dots between phytochemistry and pharmacology by linking the metabolites they found (like trimethylellagic acid) to the cytotoxic effects they observed. This helps address the lack of detailed info on how the metabolites in this plant part relate to their antiproliferative activity.
By identifying 63 metabolites, including 19 new ellagitannins and 7 anthraquinones, this work is expanding what we know about the phytochemicals in K. africana. The cytotoxic evaluation against six cancer models (like TNBC and CTCL) is also new, since most previous studies tested fruit extracts or isolated compounds (like lapachol). The IC50 values they found for the stem bark extract (4–30 µg/mL) are pretty strong compared to what’s been reported before (like 42–55 µg/mL for fruit extracts against Jeg-3 cells). By combining the UHPLC-HRMS data with the bioactivity results, they’re getting a comprehensive view of the therapeutic potential of the stem bark.
Overall, it is a good manuscript. However, the current version is not suitable for acceptance for publication. The following comments or questions may help improve the manuscript.
Reviewer: In metabolite quantification experiments, relative abundances (e.g., 8.82% abundance of dimethylellagic acid isomers) lack context.
Response: Thank you for your note. As requested, the sentence has been modified for better clarity.
Reviewer: The mechanisms linking specific metabolites (e.g., hydroxytetramethoxydihydroanthraquinone-O-hexoside) to cytotoxicity remain speculative and need to be discussed and strengthened.
Response: Thank you for the recommendation. A further discussion on the biological activity of these derivatives has been provided in the revised text.
Reviewer: Details in the figures and tables need to be strengthened.
**Table 1**: The “Confidence” column lacks clarity—define levels (e.g., Level 1: identified via standard) in the caption. Include retention time variability for isomers (e.g., punicalin α/β).
Response: Thank you for the recommendation. Confidence levels have been added under Table 1. Each isomer has an individual retention time, and tR variability is not relevant in the case of LC-HRMS.
Reviewer: **Figures 2–5**: MS/MS spectra are illustrative but lack high-resolution labels (e.g., *m/z* values). Annotate key fragments directly on spectra for readability.
Response: Thank you for the comment. The m/z values are provided, along with the chemical formulas of the compounds.
Reviewer: **Figure 1**: The TIC should align peak numbers with Table 1 entries (e.g., peak 44 = trimethylellagic acid).
Response: According to Reviewer`s 2 recommendation, Figure 1 was removed.
Reviewer: **Data quality**: High-resolution MS data (ppm errors <5) and dose-response curves (R2 values) support robustness.
Response: Thank you for the comment.
Reviewer: Some text formats require attention, such as IC50 in line 438.
Response: Thank you for noticing. The requested correction has been made.
Reviewer: The brand and manufacturer information of the MTT reagent needs to be indicated.
Response: A further clarification has been added. The MTT reagent has been purchased from Sigma-Aldrich (No.: M2128).
Reviewer “5. Conclusions” shold be “4. Conclusions”.
Response: Thank you for noticing. The correction has been made.
Round 2
Reviewer 1 Report
Comments and Suggestions for Authors
The authors have addressed the points I raised
Author Response
Reviewer 1
Reviewer: The authors have addressed the points I raised
Response: Thank you for your comment.
Reviewer 2 Report
Comments and Suggestions for Authors
In this revised version the authors address most of the remarks/suggestions;
Attention, please, must be taken in the discussion of the results based on IC50/ SI - it is still lacking a discussion regarding the IC50/SI values - I suggest statistical analysis between the different cell lines for the same cancer (normal and cancerigenous) and only then we can say if there is a strong activity or not; moreover, there are references that indicate a range of concentrations for an extract to be classified as a good/moderate/bad cytotoxic agent; as for SI, the same principles is applied - there are references that state a value of SI for a "good" potential anticancer agent;
For these methods/discussion aspects I maintain the "major revision" analysis
Author Response
Reviewer 2
Reviewer: Attention, please, must be taken in the discussion of the results based on IC50/ SI - it is still lacking a discussion regarding the IC50/SI values - I suggest statistical analysis between the different cell lines for the same cancer (normal and cancerigenous) and only then we can say if there is a strong activity or not; moreover, there are references that indicate a range of concentrations for an extract to be classified as a good/moderate/bad cytotoxic agent; as for SI, the same principles is applied - there are references that state a value of SI for a "good" potential anticancer agent;
For these methods/discussion aspects I maintain the "major revision" analysis
Response: Thank you for your recommendation. The following sentences were added to the Discussion: “Тhe primary data from the conducted cytotoxicity screenings testify to the extremely high potential of the K. africana extract as an antineoplastic remedy of natural origin. Calculated IC50 values for all tumor models ranged in the low µg/mL range, qualifying it as a highly potent plant extract, matching or far exceeding the in vitro activity of some of the most toxic plant sources, including Catharanthus roseus, Annona muricata and Curcuma longa Which is even more valuable in practical terms, the studied extract showed explicit selectivity in its cytotoxic action towards all malignant cell types, with extremely favorable selectivity indices invariably higher than 7-8 (a SI of at least 2 is considered as merely favorable). Moreover, the observed lack of toxicity towards healthy HEK-293 cells is well in line with the results of our previous study evaluating the in vivo effects of the same extract on Lewis lung carcinoma (LLC) bearing mice at far greater exposure levels. The established cytotoxicity profile of the K. africana extract, highly biased toward malignantly transformed but not normal cells, suggests specific modulation of defined molecular tumor targets, which will be the subject of future investigation, aiming to further elucidate the relationship between the phytochemical composition and anti-tumor activity of the extract.”
Round 3
Reviewer 2 Report
Comments and Suggestions for Authors
The authors have addressed the topic regarding SI, but not the one regarding IC50 and did not alter the methods section (if statistical analysis is performed there must be an indicative to that in the methods section)
For these methods/discussion aspects I maintain the "major revision" analysis
Author Response
Reviewer 2
Reviewer: The authors have addressed the topic regarding SI, but not the one regarding IC50 and did not alter the methods section (if statistical analysis is performed there must be an indicative to that in the methods section)
For these methods/discussion aspects I maintain the "major revision" analysis
Response: Thank you for your recommendation. A new section 3.6. Statistical analysis was added in the Materials and methods.